# Effects of Resveratrol on Vascular Function in Retinal Ischemia-Reperfusion Injury

**DOI:** 10.3390/antiox12040853

**Published:** 2023-04-01

**Authors:** Panagiotis Chronopoulos, Caroline Manicam, Jenia Kouchek Zadeh, Panagiotis Laspas, Johanna Charlotte Unkrig, Marie Luise Göbel, Aytan Musayeva, Norbert Pfeiffer, Matthias Oelze, Andreas Daiber, Huige Li, Ning Xia, Adrian Gericke

**Affiliations:** 1Department of Ophthalmology, University Medical Center, Johannes Gutenberg University Mainz, Langenbeckstrasse 1, 55131 Mainz, Germany; 2AbbVie Germany GmbH & Co., KG, 65189 Wiesbaden, Germany; 3Laboratory of Corneal Immunology, Transplantation and Regeneration, Schepens Eye Research Institute, Massachusetts Eye and Ear, Department of Ophthalmology, Harvard Medical School, 20 Staniford St, Boston, MA 02114, USA; 4Department of Cardiology, Cardiology 1, University Medical Center, Johannes Gutenberg University, Langenbeckstrasse 1, 55131 Mainz, Germany; 5German Center for Cardiovascular Research (DZHK), Partner Site Rhine-Main, 55131 Mainz, Germany; 6Department of Pharmacology, University Medical Center, Johannes Gutenberg University Mainz, Langenbeckstrasse 1, 55131 Mainz, Germany

**Keywords:** arterioles, ischemia-reperfusion injury, reactive oxygen species, resveratrol, retina

## Abstract

Ischemia-reperfusion (I/R) events are involved in the development of various ocular pathologies, e.g., retinal artery or vein occlusion. We tested the hypothesis that resveratrol is protective against I/R injury in the murine retina. Intraocular pressure (IOP) was elevated in anaesthetized mice to 110 mm Hg for 45 min via a micropipette placed in the anterior chamber to induce ocular ischemia. In the fellow eye, which served as control, IOP was kept at a physiological level. One group received resveratrol (30 mg/kg/day p.o. once daily) starting one day before the I/R event, whereas the other group of mice received vehicle solution only. On day eight after the I/R event, mice were sacrificed and retinal wholemounts were prepared and immuno-stained using a Brn3a antibody to quantify retinal ganglion cells. Reactivity of retinal arterioles was measured in retinal vascular preparations using video microscopy. Reactive oxygen species (ROS) and nitrogen species (RNS) were quantified in ocular cryosections by dihydroethidium and anti-3-nitrotyrosine staining, respectively. Moreover, hypoxic, redox and nitric oxide synthase gene expression was quantified in retinal explants by PCR. I/R significantly diminished retinal ganglion cell number in vehicle-treated mice. Conversely, only a negligible reduction in retinal ganglion cell number was observed in resveratrol-treated mice following I/R. Endothelial function and autoregulation were markedly reduced, which was accompanied by increased ROS and RNS in retinal blood vessels of vehicle-exposed mice following I/R, whereas resveratrol preserved vascular endothelial function and autoregulation and blunted ROS and RNS formation. Moreover, resveratrol reduced I/R-induced mRNA expression for the prooxidant enzyme, nicotinamide adenine dinucleotide phosphate oxidase 2 (NOX2). Our data provide evidence that resveratrol protects from I/R-induced retinal ganglion cell loss and endothelial dysfunction in the murine retina by reducing nitro-oxidative stress possibly via suppression of NOX2 upregulation.

## 1. Introduction

Circulatory disorders of the retina and optic nerve are a frequent cause of severe visual impairment [1,2]. For example, retinal artery or vein occlusions constitute a frequent cause of severe visual impairment leading to numerous sequelae in the eye [3,4,5]. Moreover, diabetic retinopathy, which is associated with retinal microvascular occlusion and with an elevated risk for systemic vascular complications, is one of the most common reasons of blindness in industrialized countries [6]. Vascular abnormalities and disturbances in perfusion have also been reported in glaucoma [7,8]. In all these diseases, ischemia and reperfusion (I/R) events appear to contribute to some extent to tissue damage [9,10,11,12]. During these events, excessive amounts of reactive oxygen (ROS) and nitrogen species (RNS) are generated, which are critically involved in retinal tissue damage [2]. Hence, counteracting nitro-oxidative stress is a potential strategy to improve retinal neuron survival in these diseases. One potential compound that was shown to reduce nitro-oxidative stress in numerous tissues is the phytoalexin, resveratrol. The substance was found in a variety of food sources, such as grapes, raspberries, mulberries, and peanuts [13,14]. Intriguingly, resveratrol was shown to exert tissue-protective properties in a variety of disease models, including studies that tested I/R effects in the heart, brain, spinal marrow, and kidney [15,16,17,18,19,20]. In the retina, resveratrol was shown to exert antiproliferative, antioxidant, anti-inflammatory, and anti-apoptotic effects [21,22,23,24]. In rodents, administration of resveratrol attenuated I/R-induced loss of retinal function measured by electroretinography [25,26,27]. Hence, the findings suggest that resveratrol might be a promising therapeutic agent for the treatment of ischemic retinal diseases. In previous studies from our laboratory, we observed increased vascular ROS levels together with compromised vascular function following retinal I/R events [28,29]. This was accompanied by a loss of retinal neurons even after relatively short ischemic intervals [28,29]. The objective of this study was to examine the hypothesis that resveratrol prevents from I/R-induced retinal cell damage and from vascular dysfunction in the mouse retina. Furthermore, possible mechanisms of action of resveratrol were examined.

## 2. Materials and Methods

### 2.1. Animals

All experiments involving laboratory animals were conducted according to the EU Directive 2010/63/EU for animal experiments and were approved by the Animal Care Committee of Rhineland-Palatinate, Germany (approval number: 23 177-07/G 13-1-064). Experiments were performed in 5 to 6-month-old, male C57Bl/6J mice. Mice were housed under standardized conditions with a 12 h light/dark cycle, temperature of 22 ± 2 °C, humidity of 55 ± 10%, and with free access to food and tap water.

### 2.2. Induction of Ischemia-Reperfusion Injury and Application of Resveratrol

One day before induction of I/R, mice received either saline (vehicle solution) or resveratrol (Carl Roth GmbH, Karlsruhe, Germany) at 30 mg/kg body weight diluted in saline via gavage. Previous studies reported on antioxidant effects of resveratrol in the cardiovascular system of mice for doses ranging from 10 mg/kg to 100 mg/kg per day [30,31,32]. The differences in metabolic rates and pharmacokinetics between humans and lab animals should be considered for an inter-species dose extrapolation. The FDA has formulated a table of conversion factors for the convenient calculation of human-equivalent dose from animal doses, with a factor of 12.3 for mice [33]. Thus, a mouse dose of 30 mg/kg corresponds a human dose of about 2.4 mg/kg. The daily dose for an adult human weighing 60 kg would be 144 mg. Such doses are common in human studies with resveratrol [34].

Twenty-four hours later, mice received a second dose of resveratrol or vehicle solution, respectively, and were subsequently anesthetized with ketamine (100 mg/kg, i.p.) and xylazine (10 mg/kg, i.p.). Body temperature was kept constant at 37 °C using a heating pad. Ocular ischemia was produced by introducing the tip of a borosilicate glass micropipette (100 µm diameter) into the anterior chamber. By a silicon tube, the micropipette was attached to a reservoir filled with Ringer solution (Fresenius Kabi Deutschland GmbH, Bad Homburg, Germany) that was raised above the level of the mouse to increase intraocular pressure (IOP) to 110 mm Hg for 45 min. The fellow eye, which served as control, was also cannulated in the same way and maintained at an IOP of 15 mm Hg for 45 min. Ocular ischemia was considered complete when whitening of the anterior segment of the eye was observed by microscopic examination. After removal of the micropipette tip from the eye, ofloxacin ophthalmic ointment (3 mg/g, Bausch + Lomb, Berlin, Germany) was applied to the conjunctival sac. For the following seven days, mice received resveratrol (30 mg/kg body weight) or vehicle solution once daily via gavage. On day 8, mice were sacrificed for further studies.

### 2.3. Retinal Wholemounts and Cell Counting

After mice had been killed by CO_2_ exposure, the eyes were dissected and retinas were explanted from the eye globes in phosphate buffered solution (PBS, Invitrogen, Karlsruhe, Germany). Retinal wholemounts were prepared by using fine-point tweezers and Vannas scissors and placed with the inner side down on filter paper and transferred each into a 5.0 mL Eppendorf safe-lock tube (Eppendorf SE, Hamburg, Germany) containing 4% paraformaldehyde (Sigma-Aldrich GmbH, Taufkirchen, Germany) for 1 h. Next, retinal wholemounts were washed with PBS und Triton X 0.5% twice for 10 min and then stored in the solution at −80 °C for 15 min. Afterwards, wholemounts were kept at RT for 20 min and subsequently washed twice with PBS und Triton X 0.5% for 10 min. Then, the primary antibody, which was directed against the retinal ganglion cell marker Brn3a (Santa Cruz Biotechnology, Dallas, TX, USA; #sc-31984; dilution: 1:750), was applied in blocking medium (PBS, Triton X 2% and normal donkey serum 2%) for 2 h at RT. Afterwards, wholemounts were washed for 5 min in PBS and Triton X 2% followed by four times of washing in PBS and Triton X 0.5% for 10 min. After this procedure, a secondary Alexa Fluor^TM^ 568-coupled donkey anti-goat antibody (Thermo Fisher Scientific, Waltham, MA, USA; #A-11057; dilution: 1:500) was applied for 2 h at dark. Subsequently, wholemounts were washed in PBS and Triton X 2% once for 5 min followed by four washing procedures of 10 min each in PBS and Triton X 0.5% and one wash in PBS for 10 min. Next, wholemounts were transferred into a petri dish with PBS and placed on a glass slide by using a small paintbrush. Then, slides were mounted using VECTASHIELD^®^ Mounting Medium with 4′,6-diamidino-2-phenylindole (DAPI) (BIOZOL Diagnostica Vertrieb GmbH, Eching, Germany) and cover-slipped.

In each wholemount, 16 pre-defined areas, 8 central and 8 peripheral, of 150 µm × 200 µm were photographed for cell counting as previously described [35]. Cells were counted manually in each photograph by using the cell counter plug-in for ImageJ (NIH, http://rsb.info.nih.gov/ij/ (accessed on 11 March 2019)). After counting, the mean density of cells was calculated, and the total cell number for each retina was assessed by multiplying the mean density by the area of the wholemount.

### 2.4. Retinal Arteriole Reactivity Measurement

Retinal arteriole reactivity was measured in retinal vascular preparations in vitro by using video microscopy as described previously [36,37]. First, mice were sacrificed by CO_2_ inhalation, and eyes were isolated and put into cold Krebs-Henseleit buffer. Next, the ophthalmic artery was cleaned from surrounding tissue and its side branches ligated. This was followed by gentle isolation of retina. Next, the ophthalmic artery was canulated and the retina placed onto a transparent platform. Thereafter, retinal arterioles were pressurized to 50 mm Hg via a reservoir connected to the micropipette and equilibrated for 30 min at 37 °C. Subsequently, first-order retinal arterioles were imaged under brightfield conditions and concentration-response curves for the thromboxane A_2_ receptor agonist, U46619 (10^−11^ to 10^−6^ M; Cayman Chemical, Ann Arbor, MI, USA), were conducted. Next, arterioles were preconstricted to 50–70% of the initial luminal diameter by titration of U46619 and responses to the endothelium-independent nitric oxide (NO) donor, sodium nitroprusside (SNP, 10^−9^ to 10^−4^ M, Sigma-Aldrich GmbH, Taufkirchen, Germany) and to acetylcholine (10^−9^ to 10^−4^ M; Sigma-Aldrich GmbH, Taufkirchen, Germany), which is an endothelium-dependent vasodilator in mouse retinal arterioles [38], were measured.

### 2.5. Assessment of ROS and RNS Levels

Levels of ROS were determined in situ by using the fluorescent dye, dihydroethidium (DHE), as described previously [39]. DHE is converted by ROS to highly fluorescent oxidized products (e.g., 2-hydroxyethidium, which is specific for superoxide as well as ethidium, which is the unspecific oxidation product formed by hydroxyl radicals via Fenton reaction or peroxide/peroxidase reactivity). Both oxidized DHE products intercalate with the DNA to form highly fluorescent complexes that represent a general read-out of oxidative stress.

After mice had been sacrificed and their eye globes harvested, cryosections of the eye globes of 10 µm thickness were cut. After thawing, the tissue sections were immediately incubated with 1 µM of DHE (Thermo Fischer Scientific, Waltham, MA, USA) for 30 min at 37° C. Using an Eclipse TS 100 microscope (Nikon, Yurakucho, Tokyo, Japan) equipped with a DS–Fi1-U2 digital microscope camera (Nikon) and the imaging software NIS Elements (Nikon, Version 1.10 64 bit) the fluorescence (518 nm/605 nm excitation/emission) was recorded and measured in retinal cross sections by using ImageJ (NIH, http://rsb.info.nih.gov/ij/ (accessed on 11 March 2019)). For quantification of RNS levels, an anti-3-nitrotyrosine antibody was used. After tissue fixation in paraformaldehyde (4%) for 20 min followed by 3 washes in PBS, tissue sections were incubated for 5 min in hydrogen peroxide 3% to block endogenous peroxidase activity. Subsequently, tissue sections were washed in PBS twice for 5 min followed by incubation for 30 min at RT with blocking medium composed of PBS with 0.1% Triton X, 1% bovine serum albumin and 1% normal goat serum. Next, an anti-3-nitrotyrosine rabbit polyclonal antibody (Merck Chemicals GmbH, Darmstadt, Germany; #06-284; dilution: 1:300; incubation time: 2 h at RT) was applied. After washing the slides in PBS (3 × 5 min), a goat anti-rabbit IgG H&L chain specific peroxidase conjugate (Merck Chemicals GmbH, Darmstadt, Germany; #401393; dilution: 1:200, incubation time: 1 h at RT) was applied. Then, slides were washed again in PBS (3 × 5 min) and the bound antibody was visualized by a DAB substrate kit (BIOZOL Diagnostica Vertrieb GmbH, Eching, Germany; #SK-4100; incubation time: 5 min at RT). Afterwards, tissue sections were rinsed for 5 min in running tap water. Subsequently nuclei were stained with hematoxylin (AppliChem GmbH, Darmstadt, Germany; #251344.1606;) for 3 min at RT and rinsed in running tap water for 5 min. Eventually tissue sections were dehydrated in ethanol (70%, 96%, 100% and 100%, 3 min each), incubated in xylol (3 × 3 min), embedded in Eukitt^®^ (Sigma-Aldrich GmbH, Taufkirchen, Germany; #03989;) and cover-slipped. Next, slices were viewed by transmitted light microscopy (Olympus Vanox-T AH-2; Olympus Deutschland GmbH, Hamburg, Germany) and photographed with a color video camera (TK-C1381; JVC Deutschland GmbH, Bad Vilbel, Germany). Luminance was measured in retinal cross sections by using ImageJ (NIH, http://rsb.info.nih.gov/ij/ (accessed on 11 March 2019)).

### 2.6. Quantitative PCR

Messenger RNA for the hypoxic markers, hypoxia inducible factor-1α (HIF-1α) and vascular endothelial growth factor-A (VEGF-A), the prooxidant redox enzyme, nicotinamide adenine dinucleotide phosphate oxidase 2 (NOX2), the antioxidant redox enzymes, glutathione peroxidase 1 (GPX1), hemoxygenase-1 (HO-1), superoxide dismutase (SOD)1 and SOD2, and for the nitric oxide synthase (NOS) isoforms, eNOS, iNOS and nNOS, was quantified in the retina by quantitative PCR (qPCR). After mice had been killed by CO_2_ inhalation, the eyes were immediately excised and transferred into cooled PBS (Invitrogen, Karlsruhe, Germany). The retina was then immediately isolated by Vannas scissors and fine-point tweezers, transferred into a 1.5 mL plastic tube, rapidly frozen in liquid nitrogen and stored at −80 °C. Within 3 months, the stored tissue samples were homogenized (FastPrep; MP Biomedicals, Illkirch, France), and the expression of genes was measured by SYBR Green-based quantitative real-time PCR, as previously reported [40]. RNA was isolated using peqGOLD TriFast™ (PEQLAB), and cDNA was generated with the High Capacity cDNA Reverse Transcription Kit (Applied Biosystems, Darmstadt, Germany). Real-time PCR reactions were performed on a StepOnePlus™ Real-Time PCR System (Applied Biosystems) using SYBR^®^ Green JumpStart™ Taq ReadyMix™ (Sig-ma-Aldrich) and 20 ng cDNA. Relative mRNA levels of target genes were quantified using comparative threshold (CT) normalized to the TATA-binding protein (TBP) housekeeping gene. Messenger RNA expression is presented as the fold-change to the normotensive control eye (vehicle-treated mice). The PCR primer sequences are listed in Table 1.

### 2.7. Quantification of NOX2 Expression

Sagittal cryosections (10 μm thickness) of eye globes were fixed in 4% paraformaldehyde solution for 20 min. Afterwards, slides were rinsed with PBS and incubated at RT with blocking solution containing 0.1% Triton-X-100 and 0.1% bovine serum albumin for 30 min. Next, tissue sections were incubated with a rabbit polyclonal antibody directed against NOX2 (LSBio, Seattle, WA, USA; #B12365; dilution 1:200, incubation time: 2 h at RT). Following incubation, each slide was washed in PBS three times for 5 min and incubated with a Rhodamine Red-X-coupled goat anti-rabbit polyclonal antibody (Dianova GmbH, Hamburg, Germany; #111-295-003; dilution 1:200, incubation time: 1 h at RT). Negative control sections were incubated with blocking media and the secondary antibody only. After the slides had been washed in PBS three times for 5 min, they were mounted using VECTASHIELD^®^ Mounting Medium with 4′,6-diamidino-2-phenylindole (DAPI) (BIOZOL Diagnostica Vertrieb GmbH, Eching, Germany) and cover-slipped. Next, tissue sections were viewed by using an Eclipse TS 100 microscope (Nikon, Yurakucho, Tokyo, Japan) equipped with a DS–Fi1-U2 digital microscope camera (Nikon) and the imaging software NIS Elements (Nikon, Version 1.10 64 bit) and the fluorescence was recorded and measured in retinal cross sections by using ImageJ (NIH, http://rsb.info.nih.gov/ij/ (accessed on 11 March 2019).

### 2.8. Statistical Analysis

For comparison of cell numbers, DHE, anti-3-nitrotyrosin, and anti-NOX2 staining intensity, one-way ANOVA and the Tukey’s multiple comparisons test were used, since data were normally distributed. Since mRNA expression values were not normally distributed, the Kruskal–Wallis test and the Dunn’s multiple comparisons test were used. Vasoconstrictor responses to U46619 and responses to perfusion pressure changes are reported as percent change in luminal diameter from resting diameter, while responses to acetylcholine and SNP are reported as percent change in luminal diameter from preconstricted diameter. Comparison between concentration responses was made using two-way ANOVA for repeated measurements and the Tukey’s multiple comparisons test. Results of normally distributed data are presented as mean ± SE, those of not normally distributed data as boxplots with minimum and maximum values. The level of significance was set at 0.05 and *n* represents the number of mice per group.

## 3. Results

### 3.1. Number of Cells in the Retinal Ganglion Cell Layer

Total cell number (DAPI-positive cells) in the ganglion cell layer was 123,340 ± 5758 and 93,595 ± 6398 in retinas from vehicle-treated and from I/R + vehicle-treated eyes, respectively (*p* < 0.01). This constitutes a reduction by ≈24% following I/R. In contrast, I/R induced only a marginal reduction in total cell number in resveratrol-treated mice (122,788 ± 6010 versus 117,339 ± 6269 cells, resveratrol versus I/R + resveratrol, *p* > 0.05) (Figure 1A–E). Cells in the retinal ganglion cell layer comprise vascular endothelial cells, glial cells and a large number of neurons [41]. Of note, ganglion cells, which transmit visual information from the retina via the optic nerve to several regions of the brain, account for only about half of the cells in the retinal ganglion cell layer [42]. By using an antibody against Brn3a, a specific retinal ganglion cell marker, we identified and calculated the number of retinal ganglion cells. Remarkably, I/R reduced the number of retinal ganglion cells in vehicle-treated mice. The calculated ganglion cell number was 61,622 ± 3209 and 40,080 ± 3565 cells in vehicle-treated versus I/R + vehicle-treated eyes (*p* < 0.001), which is a reduction by ≈35% following I/R. In contrast, I/R had only a negligible effect on retinal ganglion cells in resveratrol-treated mice (59,959 ± 3544 and 55,575 ± 3502 cells, resveratrol versus I/R + resveratrol, *p* > 0.05) (Figure 1F).

### 3.2. Responses of Retinal Arterioles

U46619 (10^−11^–10^−6^ M) produced concentration-dependent vasoconstriction responses of retinal arterioles that were similar in all groups (Figure 2A). Likewise, endothelium-independent vasodilation induced by SNP (10^−9^–10^−4^ M) was similar in all four groups (Figure 2B). In contrast, acetylcholine-induced (10^−9^–10^−4^ M) vasodilation was greatly reduced in arterioles from eyes exposed to I/R and vehicle (Figure 2C). Of note, resveratrol prevented endothelial dysfunction following I/R (Figure 2C). In arterioles from vehicle-treated (controls) and from resveratrol-treated eyes, stepwise increases in perfusion pressure produced pressure-dependent vasoconstriction in the pressure range between 40 mmHg and 80 mmHg, suggesting intact autoregulation (Figure 2D). In contrast, arterioles exposed to I/R and vehicle dilated with increasing pressures, indicative of abolished autoregulation (Figure 2D). Remarkably, resveratrol treatment retained pressure-dependent vasoconstriction responses following I/R (Figure 2D).

### 3.3. ROS Levels in the Retina

DHE staining in retinal cross sections revealed markedly increased fluorescence in retinal blood vessels and in the ganglion cell layer from eyes exposed to I/R from vehicle-treated mice compared to eyes not exposed to I/R (controls) (Figure 3A–F), indicative of increased ROS concentration. In contrast, fluorescent intensity the I/R resveratrol group was not significantly increased compared with the respective control group (Figure 3E,F). This per se can be anticipated as a significant difference due to the treatment. However, no difference in fluorescent intensity in blood vessels and the ganglion cell layer was observed between the I/R vehicle and the I/R resveratrol group.

Notably, DHE staining intensity did not differ between the four groups in any of the other retinal layers (Figure 3G–J).

### 3.4. RNS Levels in the Retina

Apart from ROS levels, we also determined RNS levels by staining of 3-nitrotyrosine, which forms when peroxynitrite, a cell-permeable strong oxidizing agent, reacts with the amino acid tyrosine in proteins. Hence, 3-nitrotyrosine is considered an indirect marker for peroxynitrite [43]. We found that in eyes exposed to I/R from vehicle-treated mice, staining intensity for 3-nitrotyrosine was markedly increased in blood vessels and in the ganglion cell layer (Figure 4A–F), but also to some extent in the inner and outer plexiform layers (Figure 4G,I). However, I/R did not increase 3-nitrotyrosine levels in the INL and ONL layers (Figure 4H,J). Remarkably, treatment with resveratrol partially prevented formation of 3-nitrotyrosine in blood vessels and almost completely prevented 3-nitrotyrosine formation in the retinal ganglion cell layer and the inner and outer plexiform layers (Figure 4).

### 3.5. Messenger RNA Expression

Notably, mRNA for the hypoxic markers, HIF-1α and VEGF-A, was not elevated following I/R (Figure 5A). In contrast, mRNA for the prooxidant enzyme, NOX2, was markedly elevated in the I/R vehicle group compared to the vehicle group (control), whereas expression in the I/R resveratrol group was not significantly increased compared with the respective control group (Figure 5A). NOX2 mRNA expression was only marginally elevated in resveratrol-treated mice exposed to I/R compared to the vehicle and resveratrol group (Figure 5A). No difference in expression was seen between the I/R vehicle and the I/R resveratrol group (Figure 5A). Notably, no impact of I/R was seen on mRNA expression for the three NOS isoforms (Figure 5B) or for the antioxidant enzymes, SOD1, SOD2, GPX1 and HO-1, respectively, although a trend of higher HO-1 mRNA expression levels in the I/R + vehicle group was noticed (Figure 5C).

### 3.6. NOX2 Expression in the Retina

Since quantitative PCR in whole retinal explants does not provide the information in which layer or anatomical structure a specific gene is expressed, we used an antibody directed against NOX2 to localize the enzyme within the retina and to quantify its expression on the protein level. Remarkably, anti-NOX2 immunoreactivity was increased in retinal blood vessels and in the ganglion cell layer of eyes subjected to I/R from vehicle-treated mice (Figure 6A–F), whereas no group-dependent differences in anti-NOX2 immunoreactivity were observed in the inner plexiform (IPL, Figure 6G), the inner nuclear (INL, Figure 6H), the outer plexiform (OPL, Figure 6I) and the outer nuclear layer (ONL, Figure 6J). Notably, immunoreactivity to NOX2 was not significantly elevated in eyes exposed to I/R from resveratrol-treated mice, suggesting that resveratrol prevented the I/R-induced increase in NOX2 expression.

## 4. Discussion

There are several major new findings emerging from this study. First, resveratrol prevented retinal ganglion cell loss and endothelial dysfunction in retinal arterioles following I/R, indicative of neuro- and vasculoprotective effects. Second, resveratrol reduced generation of ROS and RNS in retinal blood vessels following I/R. Third, treatment with resveratrol prevented I/R-induced increase in mRNA and protein expression for the prooxidant enzyme, NOX2.

Previous studies reported on the protective role of resveratrol on retinal ganglion cells, retinal thickness and retinal function in models of retinal I/R injury [25,27,44,45]. For example, in a mouse model of retinal I/R, resveratrol inhibited the HIF-1a/VEGF and p38/p53 pathways, while it activated the PI3K/Akt pathway, which promotes cell metabolism, proliferation and viability [27]. However, the role of retinal blood vessels has not been investigated in this regard so far. We previously reported that I/R events in the retina of mice and pigs attenuated vascular endothelial function, which went along with an increase in vascular ROS levels [28,29]. In line with our previous studies, we observed endothelial dysfunction and found elevated ROS levels in the vascular wall and in the retinal ganglion cell layer following I/R in the present study. Since vascular smooth muscle reactivity was not impaired in retinal arterioles following I/R, the compromised autoregulation appears to be an endothelium-dependent effect. Excessive ROS levels induce endothelial dysfunction via multiple mechanisms, such as uncoupling of eNOS from generation of nitric oxide or by reducing connexin and/or calcium-activated potassium channel expression resulting in impaired endothelium-derived hyperpolarizing factor (EDHF)-dependent responses [2,46,47]. In the mouse ophthalmic artery, where eNOS contributes only partially to endothelium-dependent vasodilation [48,49,50], excessive ROS formation was shown to affect various endothelial vasodilatory signaling pathways [51]. However, in retinal arterioles eNOS is the major mediator of endothelial vasodilatory responses, suggesting that eNOS uncoupling is the main mechanism via which ROS induce endothelial dysfunction [52,53].

In addition, we found elevated retinal RNS levels especially in the vascular wall and in the retinal ganglion cell layer. RNS, such as peroxynitrite and dinitrogen trioxide, are excessively generated under oxidative stress conditions [54,55]. Peroxynitrite, which is generated when superoxide rapidly combines with nitric oxide, is a reactive oxidant, which in turn further contributes to reduction in nitric oxide production and to endothelial dysfunction [2,56]. When peroxynitrite reacts with the amino acid tyrosine in target proteins, 3-nitrotyrosine forms, which is considered an indirect marker for peroxynitrite [57,58]. Peroxynitrite is involved in various retinal pathologies and has recently been suggested a novel pathomechanistic player and treatment target for glaucoma [56,59].

Importantly, resveratrol treatment attenuated the increase in ROS and RNS formation and prevented I/R-induced endothelial dysfunction and impairment of vascular autoregulation in the present study. Protective effects of resveratrol on endothelial function have been reported in other vascular beds already, such as aorta, mesenteric arteries and cerebral blood vessels [60,61,62]. Regarding the mechanisms of action, resveratrol itself as well as resveratrol-containing red wines were shown to enhance expression of eNOS in endothelial cells [63,64,65]. Apart from upregulation of eNOS expression, resveratrol was also shown to enhance enzymatic activity of eNOS via post-translational modifications, e.g., by increasing eNOS Ser-1177 phosphorylation, by inducing SIRT1-mediated deacetylation of eNOS at Lys-496 and Lys-506 or by reduction in caveolin-1 expression or its association with eNOS [66,67,68,69,70,71]. Resveratrol also prevents eNOS uncoupling, a situation in which eNOS produces superoxide under pathological conditions [31,72,73]. In cultured endothelial cells, resveratrol was reported to enhance the expression of a variety of antioxidant enzymes, including SOD1, SOD2, GPX1 and HO-1 [31,74,75,76]. The expression and activity of NOX, major ROS-producing enzymes in endothelial cells, are reduced by resveratrol [31,76,77]. Furthermore, resveratrol reduces mitochondrial superoxide production by improving mitochondrial biogenesis [78]. Based on previous studies, including our own, the prooxidant NOX2 isoform plays a critical role in retinal ROS production under hypoxic conditions or following I/R events [28,29,79,80]. Notably, NOX2 expression was shown to be upregulated in the vascular wall of retinal blood vessels under hypoxic and ischemic conditions [28,79]. Apart from I/R events, NOX2 upregulation has been associated with endothelial dysfunction in ocular blood vessels in a variety of other pathological conditions, such as elevated intraocular pressure, hypercholesterolemia and angiotensin II exposure [39,51,81,82]. Moreover, NOX2-derived ROS were reported to elevate arginase expression and activity, to decrease nitric oxide formation and to induce premature vascular endothelial cell senescence in diabetic retinopathy [83]. Apart from mediating vascular endothelial dysfunction, NOX2 was reported to be involved in mediating retinal ganglion cell death in various ocular disease models, such as traumatic optic neuropathy, glaucoma, I/R events and M_1_ muscarinic acetylcholine receptor deficiency [28,82,84,85]. Hence, blockade of NOX2 represents a promising strategy to prevent nitro-oxidative stress as well as vascular endothelial dysfunction, premature endothelial cell senescence and retinal ganglion cell death.

Notably, in the present study, resveratrol attenuated I/R-induced upregulation of NOX2 expression on the mRNA and on the protein level. Resveratrol has also previously been reported to prevent upregulation of NOX2 expression in various other disease models. For example, in a study on aged mice, resveratrol improved vascular responses and prevented upregulation of NOX enzymes, including NOX2, in the brain [86]. Likewise, in hyperhomocysteinemia-induced renal dysfunction in rats, resveratrol prevented upregulation of NOX2 and NOX4 expression in kidney tissue [87].

Apart from upregulation of NOX2 mRNA expression, we did neither observe mRNA upregulation for the hypoxic markers, HIF-1α and VEGF-A, nor for the three NOS isoforms or the antioxidant enzymes SOD1, SOD2, HO-1 or GPX1 following the I/R event. Since we harvested retinal tissue on day 8 after the I/R event for mRNA quantification, we cannot rule out the possibility that mRNA expression for some of these proteins was altered in the acute phase of the I/R event. It has previously been shown in a longitudinal in vivo study in mice that one day after the retinal I/R event ROS levels were markedly elevated whereas they were already significantly lower on the third and seventh day after the event [88]. This suggests that also regulation of some redox enzymes may have returned to normal or almost normal levels after this period. However, in the present study, we were interested in sustained changes of hypoxic, redox and NOS genes, since we previously demonstrated in several mouse models of stress-induced retinal vascular endothelial dysfunction that impairment of endothelial function persisted for weeks or even months after the stress stimulus had been stopped [29,82,89]. The present study supports the previously reported observations by suggesting that vascular endothelial recovery is not accomplished one week after the I/R event and highlights the importance of vasculo- and neuroprotective substances in retaining organ function.

## 5. Conclusions

In conclusion, resveratrol prevented retinal ganglion cells and retinal blood vessels from ischemic injury via reduction in nitro-oxidative stress. Prevention of I/R-induced upregulation of NOX2, a key enzyme involved in ROS and RNS production, is likely to be a molecular mechanism contributing to this effect of resveratrol. From a clinical point of view, resveratrol might become useful to treat various ocular conditions, in which I/R events contribute to the pathophysiology. It remains, however, to be assessed whether administration of resveratrol needs to be applied on a chronic basis to prevent from I/R events or whether administration after the I/R event will be useful. Moreover, it remains to be tested whether resveratrol may be also useful to treat chronic ischemic conditions in the retina and optic nerve. Furthermore, the optimal root of administration needs to be determined. Although resveratrol and its metabolites were detected in ocular tissues after oral administration in humans, the concentration of resveratrol was low [90]. Hence, novel formulations need to be tested to obtain a better bioavailability and targeted/prolonged delivery [91].

## Figures and Tables

**Figure 1 antioxidants-12-00853-f001:**
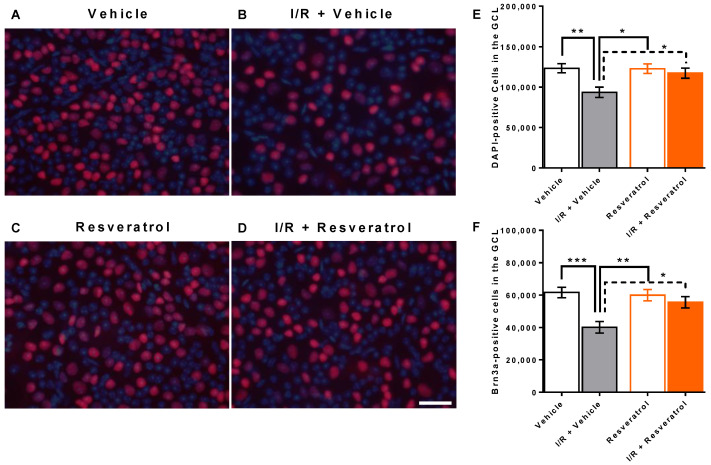
Cells in the ganglion cell layer (GCL) of the retina. (**A**–**D**) Representative pictures of cells in the ganglion cell layer stained with DAPI (blue color) to determine total cell number and with an antibody directed against Brn3a, a specific marker for retinal ganglion cells, together with an Alexa Fluor 568 secondary antibody (red color). Scale bar = 30 µm. I/R markedly reduced total cell number (**E**) and ganglion cell number in the retinal GCL (**F**) in vehicle-treated mice but not in resveratrol-treated mice. Values are expressed as means ± SE (*** *p* < 0.001; ** *p* < 0.01; * *p* < 0.05; *n* = 8 per group).

**Figure 2 antioxidants-12-00853-f002:**
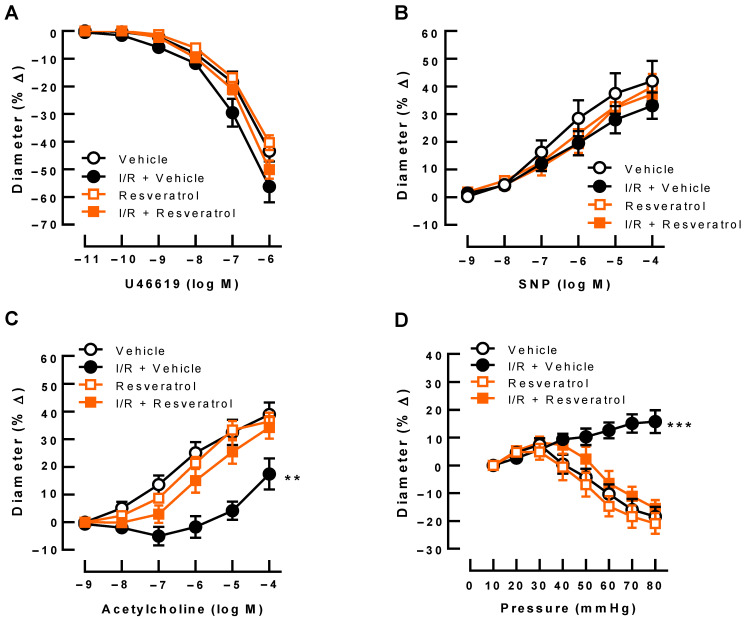
Responses of retinal arterioles to vasoactive substances. (**A**) The thromboxane A_2_ receptor agonist, U46619, produced concentration-dependent vasoconstriction responses in retinal arterioles that were similar in all groups. (**B**) Similarly, responses to the endothelium-independent vasodilator, sodium nitroprusside (SNP), did not differ between the four groups. In contrast, retinal arterioles from vehicle-treated eyes subjected to I/R displayed impaired endothelium-dependent vasodilator responses to acetylcholine (**C**) and virtually abolished responses to changes of perfusion pressure (**D**), which, however, were retained by treatment with resveratrol. Values are expressed as means ± SE (*** *p* < 0.001; ** *p* < 0.01; I/R + vehicle versus all other groups; *n* = 8 per group).

**Figure 3 antioxidants-12-00853-f003:**
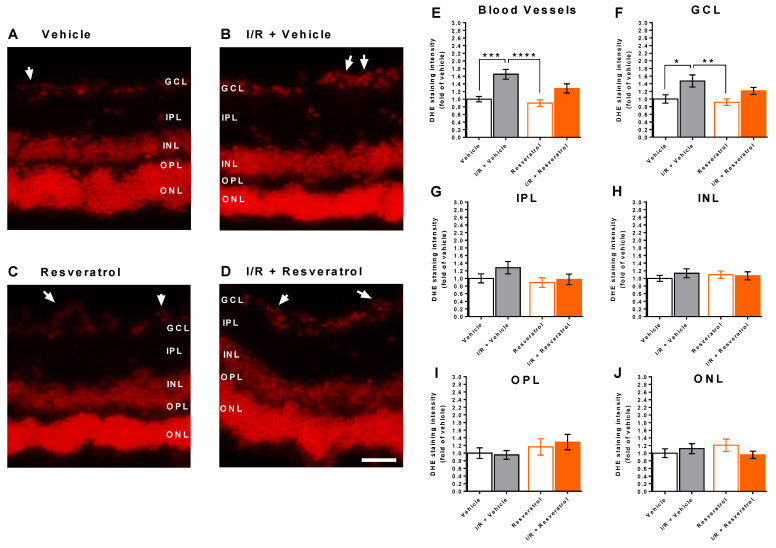
Dihydroethidium (DHE) staining in retinal cryosections. (**A**–**D**) Representative pictures of retinal cross-sections from each group. The white arrows point to the blood vessels. Scale bar = 50 µm. (**E**–**J**) DHE staining intensity was markedly increased in retinal blood vessels (**E**) and in the GCL (**F**) from I/R- and vehicle-treated eyes. In other retinal layers, no significant differences in DHE staining intensity were observed among groups (**G**–**J**). Values are expressed as means ± SE (**** *p* < 0.001; *** *p* < 0.001; ** *p* < 0.01; * *p* < 0.05; *n* = 8 per group). GCL, ganglion cell layer; IPL, inner plexiform layer; INL, inner nuclear layer; OPL, outer plexiform layer; ONL, outer nuclear layer.

**Figure 4 antioxidants-12-00853-f004:**
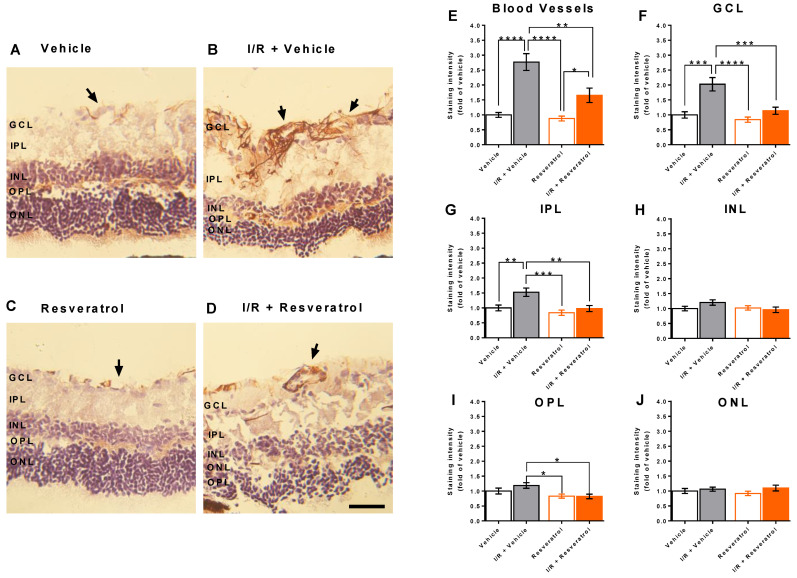
Anti-3-nitrotyrosin staining (brown color) in retinal cross-sections. (**A**–**D**) Representative pictures of retinal cross-sections from each group. The black arrows point to the blood vessels. Scale bar = 50 µm. (**E**–**J**) Anti-3-nitrotyrosin staining intensity was markedly increased in retinal blood vessels (**E**), in the GCL (**F**), in the IPL (**G**) and in the OPL (**I**) from I/R- and vehicle-treated eyes. Remarkably, resveratrol-treatment prevented the I/R-induced increase in anti-3-nitrotyrosin reactivity. In the INL (**H**) and in the ONL (**J**) no marked differences in anti-3-nitrotyrosin staining intensity were seen between groups. Values are expressed as the means ± SE (**** *p* < 0.001; *** *p* < 0.001; ** *p* < 0.01; * *p* < 0.05; *n* = 8 per group). GCL, ganglion cell layer; IPL, inner plexiform layer; INL, inner nuclear layer; OPL, outer plexiform layer; ONL, outer nuclear layer.

**Figure 5 antioxidants-12-00853-f005:**
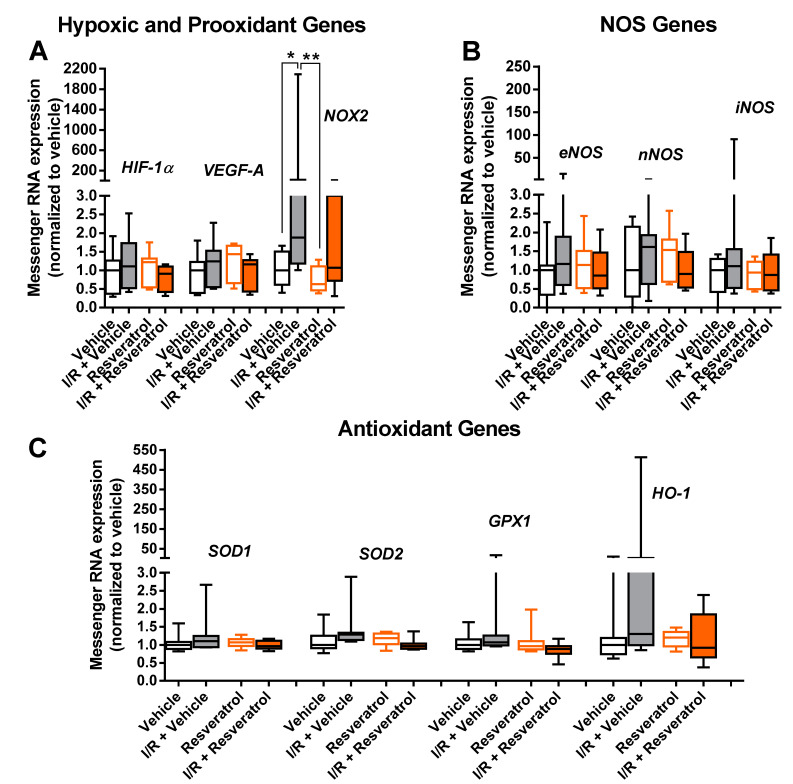
Messenger RNA expression for the hypoxic markers, HIF-1α and VEGF-A (**A**), the prooxidant enzyme, NOX2 (**A**), the three NOS isoforms, eNOS, nNOS and iNOS (**B**), and for the antioxidant enzymes, SOD1, SOD2, GPX1 and HO-1) (**C**). In eyes treated with vehicle only, I/R induced upregulation of NOX2 expression, which was not observed in resveratrol-treated eyes. Data are presented as boxplots with minimum and maximum values (** *p*< 0.01; * *p* < 0.05; *n* = 8 per group).

**Figure 6 antioxidants-12-00853-f006:**
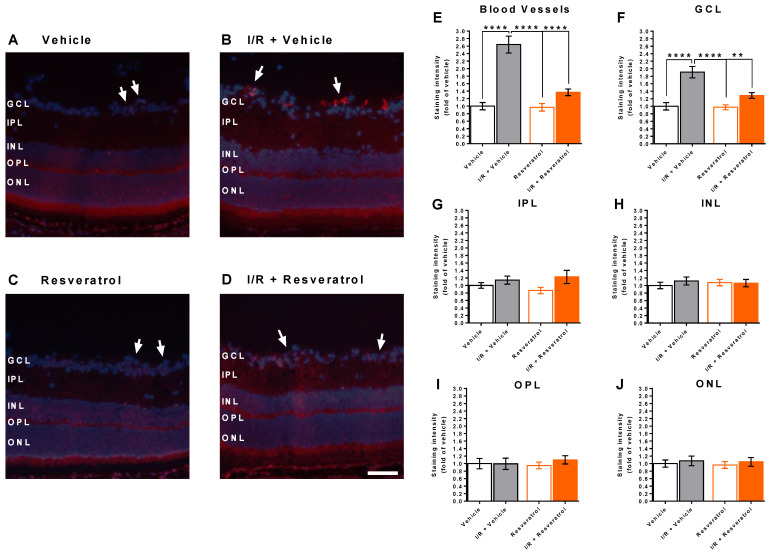
Anti NOX2 staining (red color) in retinal cross-sections. Cell nuclei were visualized with DAPI (blue). (**A**–**D**) Representative pictures of retinal cross-sections from each group. The white arrows point to the blood vessels. Scale bar = 50 µm. (**E**–**J**) Anti-NOX2 staining intensity was increased in retinal blood vessels (**E**) and in the GCL (**F**) from I/R-exposed eyes of vehicle-treated mice. Of note, resveratrol-treatment prevented the I/R-induced increase in anti-NOX2 immunoreactivity (**E**,**F**). In the IPL (**G**), INL (**H**), OPL (**I**) and in the ONL (**J**) no marked differences in anti-NOX2 staining intensity were observed between the groups. Values are expressed as the means ± SE (**** *p* < 0.001; ** *p* < 0.01; *n* = 8 per group). GCL, ganglion cell layer; IPL, inner plexiform layer; INL, inner nuclear layer; OPL, outer plexiform layer; ONL, outer nuclear layer.

**Table 1 antioxidants-12-00853-t001:** Primer sequences used for quantitative PCR analysis.

Gene	Accession Number	Forward	Reverse
*HIF-1α*	NM_010431	TCATCAGTTGCCACTTCCCCAC	CCGTCATCTGTTAGCACCATCAC
*VEGF-A*	NM_001025250.3	ACTTGTGTTGGGAGGAGGATGTC	AATGGGTTTGTCGTGTTTCTGG
*NOX2*	NM_007807.2	CCAACTGGGATAACGAGTTCA	GAGAGTTTCAGCCAAGGCTTC
*eNOS*	NM_008713	CCTTCCGCTACCAGCCAGA	CAGAGATCTTCACTGCATTGGCTA
*iNOS*	NM_010927	CAGCTGGGCTGTACAAACCTT	CATTGGAAGTGAAGCGTTTCG
*nNOS*	NM_008712	TCCACCTGCCTCGAAACC	TTGTCGCTGTTGCCAAAAAC
*GPX1*	NM_008160	CCCGTGCGCAGGTACAG	GGGACAGCAGGGTTTCTATGTC
*HO-1*	NM_010442	GGTGATGCTGACAGAGGAACAC	TAGCAGGCCTCTGACGAAGTG
*SOD1*	NM_011434.1	CCAGTGCAGGACCTCATTTTAAT	TCTCCAACATGCCTCTCTTCATC
*SOD2*	NM_013671	CCTGCTCTAATCAGGACCCATT	CGTGCTCCCACACGTCAAT
*TBP*	NM_013684	CTTCGTGCAAGAAATGCTGAAT	CAGTTGTCCGTGGCTCTCTTATT

## Data Availability

Data are all included within this article. Raw data are available from the corresponding author upon request.

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
