# Peer review of "Effects of Resveratrol on Vascular Function in Retinal Ischemia-Reperfusion Injury"

_antioxidants, 2023, doi:10.3390/antiox12040853_

Round 1

Reviewer 1 Report

This study examined the protective effects of resveratrol administration on retinal ganglion cells and retinal vascular endothelial cells in an ischemia-reperfusion (I/R) model of the retina. Several questions are raised throughout the study and are listed below.

1. Ischemia-reperfusion is a model of acute ischemic injury, and chronic ischemia is known to be involved in the actual clinical conditions described in the Introduction, such as diabetic retinopathy, age-related macular degeneration, and glaucoma. The extent to which the present ischemia-reperfusion model is involved in such chronic ischemia or circulatory disturbance requires further explanation. Perhaps this is also a limitation of this study. 

2. Please explain why the dose of resveratrol used in this study was 30 mg/kg/day. If 30 mg/kg is simply applied to an adult human weighing 60 kg, the dose would be 1.8 g/60 kg/day. Since the acceptable daily intake (ADI) for adults has been reported to be 450 mg/60 kg/day in the past, this volume is a considerably higher dose than the ADI. Please explain the significance of the experiment with such a dose. 

3. Related to the previous section, I think it is fundamentally necessary to conduct experiments not only with a single dose but also with multiple doses to see if a dose-dependent effect can be observed in order to make the study more scientifically persuasive.

4. There are not always enough statistically significant differences to support the authors' conclusions. For example, Fig. 3 shows no significant difference in ROS level between the I/R + resveratrol group and the I/R + vehicle group, and Fig. 5 shows no significant difference in NOX 2 mRNA levels between the I/R + resveratrol group and the I/R + vehicle group. These results suggest that the molecular mechanism of the antioxidant effect of resveratrol, as described by the authors in the Discussion, is not well established.

Author Response

  • Ischemia-reperfusion is a model of acute ischemic injury, and chronic ischemia is known to be involved in the actual clinical conditions described in the Introduction, such as diabetic retinopathy, age-related macular degeneration, and glaucoma. The extent to which the present ischemia-reperfusion model is involved in such chronic ischemia or circulatory disturbance requires further explanation. Perhaps this is also a limitation of this study.

To 1.) We agree with the reviewer that our model is a model of acute ischemic injury. Although diabetic retinopathy, age related macular degeneration and glaucoma differ in their pathophysiology and their pattern of tissue damage, repeated I/R events have been suggested to contribute to all these diseases. Our model imitates in the best possible way the contribution of ischemic conditions that occur in these diseases. We added a sentence in the conclusion staining the following “Moreover, it remains to be tested whether resveratrol may also be useful to treat chronic ischemic conditions in the retina and optic nerve.” (lines 490-492).

  • Please explain why the dose of resveratrol used in this study was 30 mg/kg/day. If 30 mg/kg is simply applied to an adult human weighing 60 kg, the dose would be 1.8 g/60 kg/day. Since the acceptable daily intake (ADI) for adults has been reported to be 450 mg/60 kg/day in the past, this volume is a considerably higher dose than the ADI. Please explain the significance of the experiment with such a dose. 

To 2.) The differences in metabolic rates and pharmacokinetics between humans and lab animals should be considered for an inter-species dose extrapolation. The FDA has formulated a table of conversion factors for the convenient calculation of human-equivalent dose from animal doses, with a factor of 12.3 for mice (PMID: 19508398). Thus, a mouse dose of 30 mg/kg corresponds a human dose of about 2.4 mg/kg. The daily dose for an adult human weighing 60 kg would be 144 mg. This dose was used in human studies with resveratrol (e.g. 150 mg in PMID: 22055504). We added a comment on this issue in the Methods section (lines 88 to 95).

  • Related to the previous section, I think it is fundamentally necessary to conduct experiments not only with a single dose but also with multiple doses to see if a dose-dependent effect can be observed in order to make the study more scientifically persuasive.

To 3.) We fully agree that a dose-response relationship study would be a very good idea. However, previous studies reported on antioxidant effects in the cardiovascular system of mice for various doses of resveratrol ranging from 10 mg/kg to 100 mg/kg/day PMID: 18261796 and PMID: 21530968, PMID: 36562913 PMID: 2061062. Accordingly, we used the 30 mg/kg dose in the present study, since usually doses in rodents are anyhow higher than the human ones (also due to the higher metabolic rate and turnover of drugs). We have added a sentence of explanation of the applied dose in the Methods section. As it is almost impossible to obtain approval for animal experiments that were previously published (e.g., using different doses when this was previously done), we did not apply for such dose-response relationship studies for the present project.

  • There are not always enough statistically significant differences to support the authors' conclusions. For example, Fig. 3 shows no significant difference in ROS level between the I/R + resveratrol group and the I/R + vehicle group, and Fig. 5 shows no significant difference in NOX 2 mRNA levels between the I/R + resveratrol group and the I/R + vehicle group. These results suggest that the molecular mechanism of the antioxidant effect of resveratrol, as described by the authors in the Discussion, is not well established.

To 4.) In Figure 3E we show that the DHE staining is significantly increased in the I/R vehicle group as compared to the control, whereas the I/R resveratrol group is not significantly increased as compared with the respective control group. This per se can be anticipated as a significant difference due to the treatment. The same considers Figure 5A: Nox2 expression is significantly increased in the I/R vehicle group as compared to the control, whereas the I/R resveratrol group is not significantly increased as compared with the respective control group. In addition, we show in figure 4 that 3-nitrotyrosine levels, a footprint of peroxynitrite in vivo formation and nitro-oxidative stress, is significantly different between the I/R vehicle and I/R resveratrol group. To localize NOX2 within the retina and to quantify its expression on the protein level, we conducted additional immunostainings. The results are presented in Figure 6.

Reviewer 2 Report

The research article by Dr Chronopoulos et al., entitled “Effects of Resveratrol on Vascular Function in Retinal Ischemia-Reperfusion Injury”, investigates the protective effects of resveratrol on the ischemic/reperfusion injury in the murine retina.

Ocular ischemia was induced by elevating intraocular pressure (IOP) to 110 mmHg for 45 min. Resveratrol was administered once daily for seven days at a dose of 30 mg/kg/day p.o., starting one day before the I/R event. Mice were sacrificed 8 days after the I/R event. Retinal wholemounts were prepared and analyzed in order to measure: the number of retinal ganglion cells, retinal arteriole responses by videomicroscopy, reactive oxygen species (ROS) and nitrogen species (RNS), hypoxic, redox and nitric oxide synthase gene expression.

The Authors found that resveratrol protects retina from the I/R-induced ganglion cell loss, reduction of vascular endothelial function and autoregulation, increased ROS and RNS in retinal blood vessels end reduces the expression of the prooxidant enzyme, nicotinamide adenine dinucleotide phosphate oxidase 2 (NOX2). This latter effect was supposed to be involved in the protective mechanism of resveratrol against the I/R-induced retinal damage.

The study is conceived with a solid rational and it was carried out with appropriate methods and procedures. The results obtained support the Authors’ conclusions.

I have only a few minor points to submit to the Authors.

Methods

1.    The number of mice used for each experimental approach should be reported.

2.       Indicate how the unique dose of resveratrol used for the treatment has been chosen.

3.       Is there a specific reason for excluding the retinal pigment epithelium form the analysis of the present study? Please, argument also in the discussion.

Results and Discussion

4.         It is impressive that resveratrol, though exerting a protective function against the ganglion cell loss and retinal arteriolar dysfunctions induced by I/R, does not preserve the anatomical retinal integrity, as shown in FIGs 3 and 4. How the Authors explain this?

Author Response

  • The number of mice used for each experimental approach should be reported.

To 1.) In each figure legend we write that we used 8 mice per group.

  • Indicate how the unique dose of resveratrol used for the treatment has been chosen.

To 2.) Previous studies reported on antioxidant effects in the cardiovascular system of mice for various doses of resveratrol ranging from 10 mg/kg to 100 mg/kg/day (PMID: 18261796, 21530968, 36562913 and 2061062). Accordingly, we used the 30 mg/kg dose in the present study. We have added a sentence of explanation of the applied dose in the Methods section (lines 88 to 95).

  • Is there a specific reason for excluding the retinal pigment epithelium form the analysis of the present study? Please, argument also in the discussion.

To 3.) Our goal was to test the effects of resveratrol on I/R injury in the retina, especially in the retinal ganglion cell layer, and the role of retinal blood vessels. The retinal pigment epithelium, which is supplied by nutrients and oxygen primarily from the choroid, was not the focus of our study. Moreover, the RPE is heavily loaded with pigment, which has an impact on the analysis of ROS and RNS expression using immunohistochemical and immunofluorescence techniques.

  • It is impressive that resveratrol, though exerting a protective function against the ganglion cell loss and retinal arteriolar dysfunctions induced by I/R, does not preserve the anatomical retinal integrity, as shown in FIGs 3 and 4. How the Authors explain this?

To 4.) The anatomical picture in cryosections of snap frozen tissue can be variable. Therefore, a judgement regarding the effects of resveratrol should be done with caution. However, we used cryopreservation, since we wanted to quantify reactive oxygen species by DHE.

Reviewer 3 Report

In this manuscript, the authors investigated the effect of resveratrol on vascular function. The authors showed that ischemia-reperfusion (I/R) reduced cell number using DAPI staining and a specific marker (anti-Brn3a) in the ganglion cell layer; however, resveratrol rescued these reductions. In addition, resveratrol prevented the reduction of acetylcholine-induced vasodilation and preserved pressure-dependent vasoconstrictor responses after I/R. The authors showed that levels of reactive oxygen species (ROS) and reactive nitrogen species (RNS) were affected by resveratrol. The methods used here are pragmatically accurate and the manuscript contains original information on resveratrol. The following points should be clarified.

1. In Figure 5, the authors excised whole eyes and subjected them to real-time PCR. However, they examined the levels of ROS and RNS using tissue sections. I recommend that the authors use in situ hybridization or immunofluorescence staining to detect the expression of pro- and antioxidant enzymes.

2. The authors should explain DHE in the text.

Author Response

  • In Figure 5, the authors excised whole eyes and subjected them to real-time PCR. However, they examined the levels of ROS and RNS using tissue sections. I recommend that the authors use in situ hybridization or immunofluorescence staining to detect the expression of pro- and antioxidant enzymes.

To 1.) According to the suggestion of the reviewer, we localized NOX2 within the retina and quantified its expression based on immunofluorescence microscopy. The results are presented in Figure 6.

     2.) The authors should explain DHE in the text.

To 2.) DHE is the abbreviation for “dihydroethidium”, which was already introduced in the Methods section 2.5 “Assessment of ROS and RNS levels”. In the legend to figure 3, the DHE abbreviation was also explained. We have now added a brief explanation on how ROS can be measured by DHE staining in the Methods section. It reads as follows: “DHE is converted by ROS to highly fluorescent oxidized products (e.g. 2-hydroxyethidium, which is specific for superoxide as well as ethidium, which is the unspecific oxidation product formed by hydroxyl radicals via Fenton reaction or peroxide/peroxidase reactivity). Both oxidized DHE products intercalate with the DNA to form highly fluorescent complexes that represent a general read-out of oxidative stress.” (lines 166-171).

Reviewer 4 Report

The manuscript submitted by Chronopoulos and coworkers’ fits very well within the topic of the journal. The authors performed a very interesting, well designed and complex in vivo study regarding the effect of resveratrol (administered orally) on vascular function at retinal level, after an ischemia-reperfusion injury induced by elevated intraocular pressure. Beside the direct effect of ischemia on the number of viable cells (mainly on retinal ganglion cells), authors determined the level of ROS and RNS, the expression of the major enzymes related to oxidative stress (either prooxidant or antioxidant) and the hypoxia markers by qPCR. These were accompanied by the evaluation of endothelial function and autoregulation of retinal arterioles by measuring the reactivity to U46619, SNP and acetylcholine. All these parameters give a comprehensive picture regarding the response of retina to high IOP induced ischemia and the beneficial effect of resveratrol in this experimental model.

Maybe the results regarding the expression/activity of antioxidant enzyme, as well as those of VEGF-A and HIF-1alpha, would be different if assessed earlier after the injury (as reported by Ji, ref. 23) but this does not diminish the scientific value of the results obtained.

The work is written very clearly and it was a real pleasure to read it.

I only have a minor comment. It is known that resveratrol has a very low bioavailability, it is rapidly metabolized and, most of It is excreted. Maybe a short comment on this issue (e.g. its accumulation in ocular tissue) would be appropriate

Author Response

  • I only have a minor comment. It is known that resveratrol has a very low bioavailability, it is rapidly metabolized and, most of It is excreted. Maybe a short comment on this issue (e.g. its accumulation in ocular tissue) would be appropriate.

To 1.) We thank the reviewer for this comment. We added a few sentences on that issue in the conclusion (lines 492-496).

Round 2

Reviewer 1 Report

Although the revision of the article has improved some aspects, there are still some points that appear to be scientifically and medically inaccurate.

1. Introduction

 In the introduction, the authors state that "In all these diseases, ischemia and reperfusion (I/R) events appear to critically contribute to tissue damage”. However, the articles cited by the authors (refs. 3, 7 and 8) lack evidence that I/R is involved in the pathogenesis of diabetic retinopathy (DR), AMD, and glaucoma (POAG). Although retinal artery or vein occlusion (CRAO, BRAO, CRVO, BRVO) are all pathologies caused by acute ischemia of the retina, and thus I/R may be involved, DR, AMD, and POAG are all chronic diseases, each with a unique pathophysiology distinct from I/R. As a reviewer, I cannot agree with the authors that I/R is involved in DR, AMD and POAG. Rather, it would be more accurate to focus on retinal vascular occlusions and say that I/R is similar to the pathological model of retinal vascular occlusions such as CRAO, BRAO, CRVO and BRVO.

2, Statistical significance

 The authors' response to the previous review point that there was no statistically significant difference between the I/R + vehicle group and I/R + RES in Figs. 3 and 5 is understandable. However, the fact that there was no statistically significant difference between the two groups in Figs. 3 and 5 should be clearly stated in the text. I believe this is necessary to make the paper more scientifically compelling. I would also suggest that the authors add the comments that they have given in their responses to the previous review.

Author Response

  1. Introduction

In the introduction, the authors state that "In all these diseases, ischemia and reperfusion (I/R) events appear to critically contribute to tissue damage”. However, the articles cited by the authors (refs. 3, 7 and 8) lack evidence that I/R is involved in the pathogenesis of diabetic retinopathy (DR), AMD, and glaucoma (POAG). Although retinal artery or vein occlusion (CRAO, BRAO, CRVO, BRVO) are all pathologies caused by acute ischemia of the retina, and thus I/R may be involved, DR, AMD, and POAG are all chronic diseases, each with a unique pathophysiology distinct from I/R. As a reviewer, I cannot agree with the authors that I/R is involved in DR, AMD and POAG. Rather, it would be more accurate to focus on retinal vascular occlusions and say that I/R is similar to the pathological model of retinal vascular occlusions such as CRAO, BRAO, CRVO and BRVO.

Authors’ response to 1.)

We appreciate that the reviewer challenges our statement because it helps us to reconsider the cited literature and to improve our arguments. Since visualizing I/R events is difficult because of its dynamic nature, the literature provides mostly indirect evidence on this topic. However, for some diseases it is quite convincing.

In fact, reperfusion of previously nonperfused retina has been reported in the context of both the natural course of diabetic retinopathy (PMID: 6084213, 9860002) and after anti-VEGF treatment (PMID: 33077475, 24768239). Possible mechanisms of action are transient occlusions of small blood vessels by leukocyte entrapment (PMID: 28931763, 10913667), excitotoxic damage to glial cells, which may result in impaired neurovascular coupling and hypoperfusion (PMID: 26297071) or destabilization of the vascular wall augmented by VEGF secreted in the ischemic environment (PMID: 20400620). This would explain why anti-VEGF treatment improves retinal perfusion in many cases (PMID: 35017700). Moreover, ischemia and reperfusion events may occur during changes from light to dark and vice versa. At dark, the retina consumes much more oxygen than under photopic conditions, due to the high oxygen demand of rods, which are activated during dark adaptation (PMID: 9747505, 9747505). Hence, prolonged dark adaptation, which occurs each night, may constitute a recurrent I/R event in diabetic patients and is likely to contribute to the development and progression of diabetic retinopathy (PMID: 9747505, PMID: 28109737). Some indirect evidence for the hypothesis that I/R events are involved in the pathogenesis of diabetic retinopathy comes from studies reporting similarities in capillary morphology following experimentally induced I/R and diabetes (PMID: 17197555).

I/R events have also been implicated in the pathophysiology of glaucoma. On the one hand, there are these evident cases of large intraocular pressure spikes leading to cessation of ocular perfusion, as they may occur in primary angle closure, pseudoexfoliation but also sometimes in primary open angle glaucoma, which are causing recurrent or permanent ischemic events and which we mimicked with our experimental setup. On the other hand, nocturnal blood pressure dips, which can be regarded as transient ischemic events, are a significant risk factor for glaucoma progression even when the intraocular pressure is relatively low (PMID: 30853468, 29310962, 25767134). Another supporting evidence for an involvement of I/R events in some forms of glaucoma is that transient visual field impairment could be induced by cold provocation in glaucoma patients with acral vasospasm (PMID: 30741709, 12766058). Furthermore, various functional and morphological vascular abnormalities have been observed in glaucoma patients, such as vasospasm, systemic hypotension, angiographic vascular perfusion defects, and alterations in blood flow parameters, which may result in reduced vascular perfusion in the optic nerve head and/or retina (PMID: 16962364).

As for age-dependent macular degeneration, abnormal hemodynamics have been observed in the eye (PMID: 7724181, PMID: 16565395, PMID: 18172113), which is only indirect evidence for I/R events. Therefore, we delete the disease and the respective citation from the introduction according to the suggestion of the reviewer.

We modified the sentence in the introduction to “In all these diseases, ischemia and reperfusion (I/R) events appear to contribute to some extent to tissue damage.” and added some new references supporting our statements.

  1. Statistical significance

Authors’ response to 2.)

We changed the statements in the “Results” section accordingly (lines 313-319 and 361-363).

Reviewer 3 Report

The authors have addressed all points raised in my previous reports, and their answers are satisfactory.

Author Response

Thank you!